# Bird Object Detection: Dataset Construction, Model Performance Evaluation, and Model Lightweighting

**DOI:** 10.3390/ani13182924

**Published:** 2023-09-14

**Authors:** Yang Wang, Jiaogen Zhou, Caiyun Zhang, Zhaopeng Luo, Xuexue Han, Yanzhu Ji, Jihong Guan

**Affiliations:** 1Department of Computer Science and Technology, Tongji University, Shanghai 201804, China; tongji_wangyang@tongji.edu.cn (Y.W.); jhguan@tongji.edu.cn (J.G.); 2Jiangsu Province Engineering Research Center for Intelligent Monitoring and Management of Small Water Bodies, Huaiyin Normal University, Huaian 223300, China; cloudzhang2014@126.com; 3Huai’an City Zoo, Huaian 223300, China; 17761837720@163.com (Z.L.); hanxuexue0224@163.com (X.H.); 4Key Laboratory of Zoological Systematics and Evolution, Institute of Zoology, Chinese Academy of Sciences, Beijing 100101, China; jiyanzhu@ioz.ac.cn

**Keywords:** object detection, model lightweighting, adaptive localization distillation, bird monitoring, bird counting

## Abstract

**Simple Summary:**

Both the lack of bird object detection datasets and the scarcity of technical references for evaluating the performance of object detection models and model lightweighting present challenges in the application of bird object detection technology. In this study, we have not only constructed the largest known bird object detection dataset, but also compared the performance of eight mainstream detection models on bird object detection tasks and proposed a feasible approach for model lightweighting in bird object detection. Our research results not only provide more accurate and comprehensive data for the field of bird detection, but also serve as a technical reference for users in selecting appropriate object detection models.

**Abstract:**

The application of object detection technology has a positive auxiliary role in advancing the intelligence of bird recognition and enhancing the convenience of bird field surveys. However, challenges arise due to the absence of dedicated bird datasets and evaluation benchmarks. To address this, we have not only constructed the largest known bird object detection dataset, but also compared the performances of eight mainstream detection models on bird object detection tasks and proposed feasible approaches for model lightweighting in bird object detection. Our constructed bird detection dataset of GBDD1433-2023, includes 1433 globally common bird species and 148,000 manually annotated bird images. Based on this dataset, two-stage detection models like Faster R-CNN and Cascade R-CNN demonstrated superior performances, achieving a Mean Average Precision (*mAP*) of 73.7% compared to one-stage models. In addition, compared to one-stage object detection models, two-stage object detection models have a stronger robustness to variations in foreground image scaling and background interference in bird images. On bird counting tasks, the accuracy ranged between 60.8% to 77.2% for up to five birds in an image, but this decreased sharply beyond that count, suggesting limitations of object detection models in multi-bird counting tasks. Finally, we proposed an adaptive localization distillation method for one-stage lightweight object detection models that are suitable for offline deployment, which improved the performance of the relevant models. Overall, our work furnishes an enriched dataset and practice guidelines for selecting suitable bird detection models.

## 1. Introduction

Changes in bird species and numbers are important indicators of the health of bird communities and their habitats. Identification of bird species and quantification of their abundance in a target region is one of the key tasks in bird field surveys [1]. However, currently, bird field monitoring is mainly conducted manually, and there are obvious limitations in this type of data collection process. (1) Birds have large habitats that are difficult to cover comprehensively through manual observation. (2) Accurate bird identification requires personnel with bird-specific knowledge or training, which hinders the participation of the general public to some extent. (3) Both the small sizes of birds and the randomness of when and where they occur make bird monitoring costly. Leveraging computer vision and deep learning technologies holds good promise for partially replacing manual bird monitoring tasks [2], especially in stationary observation areas. This approach may effectively reduce the manual effort and resources required for bird field surveys. Deep learning techniques have been used with significant success in image classification, object detection, and other fields [3,4]. For example, convolutional neural network classification methods such as VGGNet [5], GoogleNet [6], and ResNet [7] have been applied to bird image classification [8,9]. Object detection models like Faster R-CNN [10], YOLO [11,12], and GFL [13] have been employed for animal object detection tasks. Overall, the current development of object detection models can be categorized into two main types: two-stage detectors and one-stage detectors.

Two-stage object detection models involve two distinct stages of operations, where the first stage generates sparse region proposals, and the second stage performs regression and classification operations. RCNN [14] was the pioneering method that successfully introduced deep convolutional neural networks (CNNs) to the field of object detection. It utilized low-level computer vision algorithms such as Selective Search [15] and Edge Boxes [16] to generate proposals, and then employed CNNs to extract features for training support vector machine classifiers and bounding box regressors, achieving good detection results. Building upon RCNN, SPP-Net [17] introduced spatial pyramid pooling, enabling effective feature extraction and position invariance for objects of different scales, thereby improving the accuracy and speed of object detection. Fast RCNN [18] involves spatial pyramid pooling on a shared feature map to extract features for each proposal, achieving better detection performance than SPP-Net. Faster RCNN [10] integrates the region proposal process into the Deep ConvNet, thus making the entire detector an end-to-end trainable model with higher efficiency compared to previous models. Additionally, R-FCN [19] introduced a region-based fully convolutional network to generate region-sensitive features, and it is capable of handling images of arbitrary sizes and further enhancing detection efficiency. FPN [20] introduces a top-down lateral connection architecture to generate a feature pyramid for multi-scale detection, thus improving the accuracy and robustness of object detection.

Unlike the aforementioned two-stage object detection models, one-stage object detection methods directly perform classification and regression on dense anchor boxes without generating a sparse region proposal set. Among them, YOLO [11] and SSD are the most representative models in the field of one-stage object detection. YOLO detects objects directly on dense feature maps, while SSD [21] utilizes multi-scale features to detect objects of different scales. Both methods demonstrate fast detection capabilities but tend to perform poorly in detecting small objects. Subsequently, RetinaNet [22] introduced focal loss to address the issue of extreme class imbalance in dense object detection. It effectively handles the imbalance between a large number of background samples and positive samples, thereby improving the detection performance for small objects. Refinedet [23] introduced anchor refinement modules and object detection modules to simulate cascade regression on dense anchor boxes, thereby enhancing the detection performance of one-stage methods. Additionally, Guided Anchor [24] first employed anchor-guided deformable convolutions to align the features of the Region Proposal Network (RPN), thereby improving the accuracy and robustness of one-stage object detection. EfficientDet [25] adopted EfficientNet [26] as the backbone network and introduced the Bi-directional Feature Pyramid Network (BiFPN) and the Complete Intersection over Union (CIOU) loss function to enhance the accuracy and efficiency of object detection. Overall, compared to two-stage object detection models, one-stage object detection methods exhibit better operational efficiency but relatively lower detection accuracy.

Although there are many object detection models available for bird detection tasks, the problem of bird object detection still requires further research. On one hand, there is a lack of large-scale and open-source bird detection datasets that cover a wide variety of bird species. Currently available bird datasets, such as the bird image data provided by Imagenet, treat birds as a subclass, and the waterbird dataset by Zhou et al. [9] covers over 500 species. However, these datasets are only suitable for bird image classification and are not specifically designed for object detection tasks. The COCO dataset, which serves as a benchmark for evaluating object detection model performance, includes a bird image subset that does not differentiate bird species. The CUB200-2011 [27] dataset is the first truly dedicated bird object detection dataset, containing images of 200 bird species from North America, with nearly 12,000 images. However, this dataset is limited by the fact that each image only contains a single bird, which does not meet the requirements of bird counting or multi-bird recognition tasks. Additionally, the bird species in these datasets are limited to those found in North America, thus lacking a representation of birds distributed in other parts of the world. Therefore, it is crucial to construct a large-scale bird detection dataset that encompasses a wider variety of bird species, a broader geographical distribution, and includes images with multiple birds in a single image. This is necessary to advance research in bird object detection. On the other hand, the wide range of existing object detection models has made it challenging for users to select an appropriate model. These models are often built using different backbone networks, and the differences in backbone networks can impact the detection model’s performance. Hence, it is necessary to evaluate the performance differences of current mainstream object detection models for bird detection tasks via a unified backbone network. This will aid users in making decisions when selecting suitable object detection models.

In addition to the scarcity of bird object detection datasets and the limitations in model selection, bird object detection tasks also face challenges in model size. Currently, in many bird observation sites, auxiliary devices such as wildlife cameras and infrared-triggered detectors are used in bird monitoring. These devices typically have small form factors and limited computational capabilities, making it impractical to directly embed existing large-scale, high-computational-cost object detection models for real-time bird detection. Therefore, there is a need to deploy lightweight parameter-optimized object detection models on these low-performance devices and perform dedicated hardware optimizations. Once the models are deployed, they are often not easy to replace. To continuously improve the performance of offline models, knowledge distillation techniques are typically employed to transfer the detection capabilities of the current state-of-the-art models to lightweight models. In essence, knowledge distillation involves using a high-performance large model as a teacher to guide the lightweight model in improving its learning performance. Hinton et al. [28] pioneered the field of knowledge distillation by using K–L divergence to optimize the probability distribution of class predictions between the teacher and student, thus enabling the student to learn the probability distribution information from the teacher’s classification results. Inspired by this research, many pioneering studies [29,30,31] in knowledge distillation have emerged. For example, an informational–theoretical framework [32] has been proposed to guide the student toward the teacher by maximizing the mutual information between the teacher and student networks. Subsequently, the value of inter-instance correlations as knowledge has also been demonstrated [33,34]. Furthermore, DarkRank [35] saw the introduction of a method that transfers knowledge through cross-sample similarity in metric learning. In the field of object detection, techniques such as ROI simulation distillation [36], fine-grained feature imitation [37], and localization distillation [38] have been designed to distill knowledge by focusing on the regions of interest and bounding box regression distribution, thus achieving good results. However, current object detection distillation methods still have some limitations when dealing with errors in the teacher model, as erroneous knowledge can be transferred to the student model. Therefore, it is necessary to selectively choose the knowledge from the teacher model, specifically addressing this issue so as to avoid the propagation of incorrect knowledge and to ensure a further enhancement of the effectiveness of distillation.

Motivated by the aforementioned challenges, we delved into the construction of bird detection datasets, the evaluation of object detection models’ performance, and model lightweighting. The overall flowchart of our research can be observed in the Figure A1. The main contributions of our study include the following:Addressing the lack of existing bird detection datasets, we have constructed the largest known bird detection dataset to date, encompassing 1433 common bird species worldwide.We systematically evaluated the performance of mainstream two-stage and one-stage object detection models for bird detection tasks within a unified backbone framework.Through experimentation, we have verified that existing mainstream object detection models perform poorly in bird counting tasks when there are more than five birds in the same image, and this is due to factors such as bird crowding and occlusion.We propose an adaptive object detection distillation method that avoids the propagation of erroneous knowledge, thus further improving the detection performance of lightweight detection models.

## 2. Materials and Methods

### 2.1. Construction of a Global Bird Object Detection Dataset

Currently, the commonly used open-source bird object detection dataset is CUB200-2011, which was composed by the California Institute of Technology. Although this dataset contains 200 species of birds from North America, it still suffers from limitations such as limited bird species diversity and a lack of image variability, thus making it difficult to meet the requirements of real-world bird object detection tasks, especially for tasks involving multiple bird detection, such as “one image, multiple birds”. To address this, we have constructed a global bird object detection dataset (GBDD1433-2023), and the process of the dataset is as illustrated in the Figure 1. Given the diverse and widespread distribution of bird species worldwide, as well as the availability of free-bird image resources scattered across the Internet, we utilized web crawling techniques with both the Chinese and English names of bird species as matching keywords to gather bird images. The collected bird image dataset inevitably includes quality issues such as low resolution, incomplete images, and mismatches between images and bird names, thus it requires data cleaning and processing. Firstly, we removed images with low resolution and manually filtered out cartoons, paintings, and images with incomplete bird bodies (which could lead to indistinguishable categories). Next, in collaboration with ornithology experts, we used detailed avian information from the Avibase website to verify the match between each bird image and its name. We retained the images that matched, while the mismatched ones were removed. These steps ensured the correct labeling of each bird image in the dataset. Finally, using third-party software named “Sprite Annotation”, we added bounding box annotations to all the images and saved the annotation information in the universal COCO format (which is widely used in the object detection field), thus resulting in creation the GBDD1433-2023 dataset. It is important to note that bird species with fewer than 20 images were not annotated with bounding boxes to minimize the impact of imbalanced sample quantities on model generalization. However, even though we removed bird species with fewer than 20 samples, the distribution of bird image quantities still exhibits a long-tailed pattern (Figure 2).

The constructed GBDD1433-2023 dataset covers 32 orders, 132 families, 572 genera, and 1433 species of birds, with a total of 148 k diverse and high-quality bird images manually annotated. It is currently the bird object detection dataset with the largest coverage of global bird species. This dataset will provide valuable data resources for researchers involved in fields such as bird intelligence monitoring and deep learning.

### 2.2. Constructing Test Datasets for Evaluating Bird Object Detection

To systematically evaluate the performance of existing object detection techniques in bird detection, we constructed two bird object detection performance test datasets: the multimodal bird object detection test dataset (MMBT) and the multi-object bird object detection test dataset (MOBT). In real-world applications, bird images not only have complex and diverse background scenes, but also exhibit variations in the scale of the foreground objects (birds) themselves. The MMBT dataset was designed to evaluate the sensitivity of object detectors to background noise and variations in the scale of foreground bird images. Through the use of image cropping and composition techniques, we created this dataset. During the construction of the MMBT dataset, we randomly selected high-resolution images of 200 bird species (one image per species) and generated foreground bird images at nine different pixel scales: 25 × 25, 50 × 50, 75 × 75, 100 × 100, 125 × 125, 150 × 150, 175 × 175, 185 × 185, and 200 × 200. Considering the possible scenarios where birds may appear, we selected 12 background scene images, including grasslands, tundras, shrubs, farmlands, and swamps (Figure A2), with a unified pixel scale of 224 × 224. Finally, we combined the different foreground bird images with the background scene images, resulting in a dataset of nearly 21,000 multimodal bird images (refer to the Figure 3).

Furthermore, we have constructed the dataset of MOBT to help with evaluating the capability of object detectors in determining multiple birds in images. In many bird counting tasks, bird images often exhibit challenges such as mutual occlusion and size variations, which pose significant challenges for bird detection. However, the majority of existing bird datasets consist of single bird images, making it impossible to assess the performance of models in detecting multiple birds. We have created a dataset that includes bird images with varying numbers of birds (see Figure 4). The dataset consists of a total of 1216 images, and the number of birds in each image has been manually annotated. This dataset presents a challenging task for existing object detection techniques. It is worth noting that the images in the two test datasets we constructed, as well as those in the aforementioned global bird object detection dataset, do not overlap with each other.

### 2.3. Principles of Model Selection and Overview of Selected Models

Continuous development in deep learning has led to a wide variety of object detection models. Evaluating the performance differences of these models in bird object detection tasks is helpful for users in choosing appropriate models. Considering the large number of published object detection algorithms, comparing the performances between all of the methods is beyond the scope of this paper. Therefore, we selected a subset of mainstream and representative models for performance evaluation. We set the following principles for model selection. Firstly, the selected models should have open-source code and be widely used in the industry. We did not consider models with closed-source code or models with limited references. Secondly, given the continual emergence of novel methods and technologies in the field, the chosen model must have been academically recognized as a state-of-the-art model at that time. Furthermore, to ensure fairness in the model environment configuration during performance evaluation, the selected models should be able to run on the same backbone model.

Based on these principles, for two-stage object detection methods, we have chosen three representative models: Faster RCNN [10], Cascade RCNN [39], and Libra RCNN [40]. These models have been widely adopted in the industry. For instance, Cascade RCNN, by leveraging a cascaded multi-model approach, demonstrated state-of-the-art performance at the time in which it was introduced (it was especially state-of-the-art in addressing class imbalance and hard sample imbalance issues, and it also surpassed other models, thus making it an essential reference methodology in the object detection domain. For one-stage object detection methods, we have selected five representative methods: YOLOv3 [41], RetinaNet [22], FCOS [42], ATSS [43], and GFL [13]. YOLOv3 has the advantage of efficient and fast real-time performance and is the most representative and widely applied method in the YOLO series. RetinaNet is suitable for multi-scale object detection, and it delivers state-of-the-art detection accuracy for objects of diverse scales. FCOS is specialized in small object detection and shows sensitivity to smaller-sized objects. ATSS introduces adaptive training sample selection to handle class imbalance and hard sample imbalance problems, thus enhancing the detection capability of long-tailed distribution objects. GFL captures more of the boundary information of objects, thus improving the detection performance for dense objects. Overall, the selected detection methods have their own advantages. The following is a brief introduction to the selected models.

#### 2.3.1. Two-Stage Object Detection Methods

Faster RCNN [10]: The earlier proposed Fast RCNN utilized Selective Search for Region of Interest (RoI) searching, which resulted in slower processing speed. Faster RCNN, built upon Fast RCNN, introduced the Region Proposal Network (RPN) to automatically generate RoIs, thus significantly improving the efficiency of proposal generation. Faster RCNN incorporates the RPN network to assist in generating samples. The algorithm structure is divided into two parts: first, the RPN network determines whether the candidate boxes are objects; second, the multi-task loss for classification and localization is used to determine the object type. The entire network flow can share the feature information extracted by a convolutional neural network, thus saving computational costs. It addresses the issue of slow candidate box generation in Fast RCNN, and it avoids the decline in accuracy caused by excessive candidate box extraction.

Cascade RCNN [39]: This model recognizes a significant difference in the proposals used during the training and inference stages of the object detection process. Specifically, the proposals in the training stage undergo a selection process (IOU > threshold) to ensure higher quality, while the proposals in the inference stage are not filtered and therefore have lower quality. Consequently, using a detector trained with the former to test the latter’s proposals can result in a decrease in model performance. To address this issue, Cascade RCNN proposes a training method that utilizes different IOU thresholds and alleviates the problem by cascading multiple stages of object detectors, thus further optimizing the detection results.

Libra RCNN [40]: This method addresses three imbalanced issues in object detection: the representativeness of sample selection, the sufficient utilization of extracted features, and the optimality of the object loss function. Libra RCNN incorporates three components: IOU-balanced sampling, balanced feature pyramids, and balanced L1 loss. These components respectively address the three imbalanced issues mentioned above. It effectively handles the sample distribution problem of different categories and difficulty levels in the dataset, thereby improving the detection performance for small objects and challenging samples.

#### 2.3.2. One-Stage Object Detection Methods

YOLOv3 [41]: Compared to traditional object detection methods, YOLOv3 has the capability to simultaneously predict bounding boxes and classes for multiple objects in an image during a single forward pass, making it an efficient solution for object recognition and tracking. Key features of YOLOv3 include real-time processing, multi-scale feature fusion, prior boxes and anchor boxes, and multi-scale predictions. YOLOv3 introduces the concept of prior boxes and anchor boxes, where a set of anchor boxes is predefined and adjusted based on the sizes of the objects. This allows the model to effectively detect objects of various sizes and shapes. The use of prior boxes and anchor boxes enables the model to adapt well to different object sizes, thus improving the detection performance. Additionally, YOLOv3 performs predictions on feature maps at different scales, allowing it to detect objects of different sizes and enhancing the model’s ability to make multi-scale predictions, thereby improving its generalization performance.

RetinaNet [22]: This model primarily addresses the issue of imbalanced positive and negative samples in the single-stage object detection process. In multi-stage object detection, methods such as Selective Search and Region Proposal Networks (RPN) can filter out a large number of background boxes, and then training is performed by selecting positive and negative samples. However, single-stage object detection algorithms cannot filter out these background boxes, resulting in a severe imbalance between positive and negative samples. To tackle this issue, the model introduces Focal loss, which adaptively adjusts the loss weights during training to focus more on challenging samples. Additionally, the model proposes the RetinaNet object detection framework, which effectively alleviates the imbalance in positive and negative sample selection.

FCOS [42]: FCOS introduces a creative anchor-free solution for object detection. Instead of relying on predefined anchors, FCOS predicts the distances from each pixel to the top, bottom, left, and right boundaries of the corresponding object box. Firstly, if an anchor falls within multiple object boxes, the model selects the smallest object box as the regression target. Secondly, to address the issue of predicting an excessive number of object boxes, the model proposes the concept of center-ness. It learns a center-ness score for each position, which is multiplied by the predicted class score during the non-maximum suppression process to filter out the predicted object boxes. With these strategies, FCOS enhances its ability to detect small objects.

ATSS [43]: ATSS further explores the fundamental difference between anchor-based and anchor-free methods, which lies in the difference in sample selection for positive and negative examples. To address this, ATSS proposes an adaptive training sample selection method that bridges the gap between anchor-based and anchor-free detectors. Overall, the ATSS approach improves the detection performance of one-stage detectors without introducing any additional overhead.

GFL [13]: Existing bounding box estimation methods have not taken into account the ambiguity and uncertainty present in the datasets. To address these problems, the GFL model integrates quality estimation into the class prediction vectors, thus forming a joint representation, which represents the bounding box positions with a vector that captures arbitrary distributions. The improved representation eliminates the risk of inconsistency and accurately characterizes the flexible distribution of real data. However, it includes continuous labels, which go beyond the scope of Focal Loss. Therefore, the model further proposes the Generalized Focal Loss (GFL), which extends the Focal Loss from a discrete form to a continuous form for successful optimization, thus achieving higher inference speeds.

### 2.4. The Self-Adaptive Localization Distillation Method

Knowledge distillation is a technique that utilizes a high-performance large model, known as the teacher, to guide a lightweight model, known as the student, in improving its learning performance. Knowledge distillation allows the transfer of knowledge and experience from the teacher model to the student model, thus enabling the student model to better generalize and adapt to various samples without requiring changes to its structure. Therefore, knowledge distillation is commonly used in practical applications to enhance the performance of lightweight models. Previous knowledge distillation (KD) methods [44] mainly focused on feature mimicry rather than logit mimicry, resulting in lower efficiency in extracting localization information. The Localization Distillation (LD) method, for the first time, demonstrated that logit mimicry can outperform feature mimicry, and that it can significantly improve distillation performance by effectively distilling target localization information. In the LD method, the definition of the localization distillation loss is shown in the Equation (Equation 1).
(1)$$L_{LD}^{e}=H\left(S\left(Z_{S},\tau \right),S\left(Z_{T},\tau \right)\right)$$

In the Equation (Equation 1), the distribution of bounding boxes quantifies the continuous regression range $\left[e_{\min},e_{\max}\right]$ into a uniform discrete variable $e=\left[e_{1},e_{2},\ldots ,e_{n}\right]\in R^{n}$, with *n* sub-intervals, where $e_{1}=e_{\min}$ and $e_{n}=e_{\max}$. $Z_{S}$ and $Z_{T}$ represent the *n* logits predicted by the student and teacher, respectively. $Z_{S}$ and $Z_{T}$ are the input to the generalized SoftMax function S(·,τ), which is performed to obtain their corresponding probability distributions. · represents a placeholder to indicate that data can be entered here. It is imperative to note that when τ=1, it corresponds to the standard SoftMax function. As τ approaches 0, it tends toward the Dirac distribution. On the other hand, as τ approaches infinity, it decays to a uniform distribution. Empirically, setting τ>1 serves to smooth the distribution, thereby enabling the probability distribution to encapsulate richer information [38]. Finally, the output is obtained through the cross-entropy loss H(·).

Although the LD method effectively enhances the performance of lightweight models, LD is still a knowledge distillation method. Therefore, the LD method inherently suffers from the limitation that even the best teacher model cannot provide completely accurate knowledge for the student model to learn, and that the transmission of incorrect knowledge can restrict the effectiveness of distillation. To alleviate this issue, we extended the LD method and propose a self-adaptive localization distillation (SLD) method, which reduces the transmission of erroneous knowledge. The core principle of the SLD method is to reinforce the student model’s inclination to learn from the teacher model when the teacher model accurately localizes objects, and to guide the student model to learn from ground truths when the teacher model inaccurately localizes objects. To achieve this, we introduced the confidence parameter *C* to measure the degree of accurate localization by the teacher model for the current target. The confidence parameter *C* can adjust the magnitude of the loss values generated by the student and teacher models during distillation. A higher value of *C* encourages the student model to lean toward learning from the teacher model, while a lower value does the opposite [45]. The construction of the confidence parameter involves two components: scoring by the teacher model and scoring by the student model. Firstly, we define the student confidence parameter $c_{1}$ based on the student model’s IOU score for the current bounding box. The composition of $c_{1}$ is shown in the Equation (Equation 2).
(2)$$c_{1}=\left\{\begin{array}{cc} k_{1}, & S_{iou}>S_{avg}\\ k_{2}, & S_{iou}<S_{avg} \end{array}\right.$$

In the equation, $S_{iou}$ represents the student model’s IOU score for the current bounding box, and $S_{avg}$ is the average IOU score of the student model’s predictions for all bounding boxes when trained independently. When $S_{iou}$ is smaller than the average, indicating that the student model performs poorly in learning ground truths for that particular object, we consider strengthening the weight $c_{1}$ toward learning from the teacher model, with a value of $k_{1}$. Conversely, when $S_{iou}$ is larger than the average, we weaken the weight $c_{1}$ toward learning from the teacher model for that object, with a value of $k_{2}$.

Next, we define the teacher’s confidence parameter $c_{2}$ based on the teacher model’s IOU score for the current bounding box. The construction of $c_{2}$ is given by the Equation (Equation 3):(3)$$c_{2}=\left\{\begin{array}{cc} k_{2}, & T_{iou}>T_{avg}\\ k_{1}, & T_{iou}<T_{avg} \end{array}\right.$$

In the formula, $T_{iou}$ represents the IOU score of the current bounding box predicted by the teacher model, and $T_{avg}$ is the average IOU score of all bounding boxes when the teacher model is trained independently. When $T_{iou}$ is less than the average, we consider that the teacher model does not perform well in learning from ground truths for this particular object, and the weight $c_{1}$ should be set as $k_{1}$ to encourage the student model to learn from ground truths. Conversely, when $T_{iou}$ is greater than or equal to the average, the weight $c_{1}$ should be set as $k_{2}$ to strengthen the student model’s learning from the teacher model for this object. Experimental results have shown that setting $k_{1}$ as 0.5 and $k_{2}$ as 2.0 yields the best performance for the model.

The confidence parameter *C* is composed of $c_{1}$ and $c_{2}$, as shown in Formula (Equation 4), where Mul denotes element-wise multiplication. The final Self-Adaptive Localization Distillation (SLD) loss function is obtained by multiplying the confidence parameter with the localization distillation function, as shown in the Formula (Equation 5). The overall training framework of SLD is illustrated in the Figure 5.
(4)$$C=Mul\left(c_{1},c_{2}\right),$$
(5)$$L_{SLD}=C\cdot L_{LD}^{e}.$$

## 3. Results and Analysis

### 3.1. Experimental Setup and Evaluation Metrics

In the subsequent experiments of model performance evaluation, we primarily evaluated the selected eight object detection models using two datasets: the publicly available benchmark dataset COCO and our self-built GBDD1433-2023 dataset. The COCO dataset consists of approximately 320,000 manually annotated images covering 90 common object categories, including humans, birds, vehicles, and household items. By using the COCO dataset, we can effectively evaluate the detection capabilities of the selected models for diverse objects. The GBDD1433-2023 dataset, specifically designed for bird detection, is the largest bird detection dataset known to date. This dataset enables us to evaluate the performance differences among the eight selected models in bird detection tasks, thus assisting users in selecting appropriate object detection models.

To ensure a fair comparison of different object detection models, all the object detection models in the paper were implemented based on the MMdetection framework [46]. The backbone network of all models was set to ResNet50, and similar training strategies were employed. During the model training and evaluation process, 80% of the data were used as training samples, and the remaining 20% were used as test samples. The models were trained end-to-end via the SGD optimizer, with a learning rate of 0.02 and a weight decay coefficient of 0.0001. The batch size was set to 8, and a total of 24 epochs were trained. The training was conducted on NVIDIA RTX3090Ti GPUs.

The detection performance of the object detection model was evaluated using the Mean Average Precision (*mAP*) metric. mAP represents the average area under the Precision–Recall curve for all categories at a specified Intersection over Union (IOU) threshold. IOU refers to the overlap ratio between the detection box and the ground truth box, which can be used to measure the degree of match between the detection result and the real object. Typically, the higher the *mAPx* value at a given IOU threshold *x* (ranging from 0 to 1), the better the model’s detection performance. To calculate *mAP*, one must first compute the Precision and Recall values for each category and then plot the Precision–Recall curve. For each category, the area under its Precision–Recall curve is calculated to obtain the Average Precision *(AP)* for that category, which measures the detection performance of a single category. The average of the APs for all categories results in the *mAP*, which is the mean value of *AP* for all categories. The formulas for computing Precision and Recall are as follows:(6)$$Precision=\frac{TP}{TP+FP}$$
(7)$$Recall=\frac{TP}{TP+FN}$$

Among them, *TP* (True Positives) represents the number of positive samples correctly detected, *FP* (False Positives) represents the number of negative samples incorrectly detected as positive, and *FN* (False Negatives) stands for the number of positive samples that were not detected. Next, by calculating the area under each point on the Precision–Recall curve (using the trapezoidal area method) and summing them up, we obtained the *AP*. The formula for calculating *AP* is as follows:(8)$$AP=\sum_{n}\left(R_{n}-R_{n-1}\right)P_{n}$$

Among them. $R_{n}$ and $P_{n}$ are Recall and Precision at the nth point, respectively. Finally, for calculating the overall *mAP*, we utilized the following:(9)$$mAP=\frac{1}{C}\sum_{c=1}^{C}AP_{c}$$
where *C* is the number of categories and $AP_{c}$ is the *AP* of the *c*th category. *mAP50* represents the mean average precision when the IOU threshold is 0.5, while *mAP75* represents the mean average precision when the IOU threshold is 0.75. Moreover, in subsequent model sensitivity experiments and bird counting experiments, we used the Accuracy metric (i.e., the ratio of correctly identified samples to the total number of samples) to evaluate model performance.

### 3.2. Performance Comparison of Object Detection Models based on the COCO and GBDD1433-2023 Datasets

We conducted a performance comparison of eight object detection models on the COCO dataset, and the results of this are presented in Table 1. Taking into account the *mAP*, *mAP50*, and *mAP75* evaluation metrics, the detection accuracy for the eight object detection models ranged from 36.5% to 40.3%, 55.4% to 59.5%, and 39.1% to 44.0%, respectively. Overall, the two-stage detection model, Cascade RCNN, exhibited the best performance among the eight models. The one-stage detection models, ATSS and GFL, performed slightly lower, while the other four models performed better than the RetinaNet model but lower than the top two models. The RetinaNet model exhibited the lowest performance among all the models. In terms of model size, the two-stage detection models were generally larger than the one-stage detection models, especially the Cascade RCNN model with the largest model size of 265.0 M.

Further evaluation of the abovementioned eight object detection models was conducted on the GBDD1433-2023 dataset, and the results are shown in Table 2. Compared to the detection performance on the COCO dataset, the performance of the eight object detection models were significantly improved on the GBDD1433-2023 dataset. Among them, the *mAP* values of the eight detection models ranged from 68.5% to 74.0%, with an average increase of 86.8%; the mAP50 values ranged from 90.6% to 92.4%, with an average increase of 58.7%; and the *mAP75* values ranged from 80.4% to 85.0%, with an average increase of 99.0%. This clearly demonstrates that our self-built GBDD1433-2023 dataset effectively enhances the performance of the eight detection models in bird detection tasks. In terms of detection accuracy, the three two-stage detection models performed relatively better than the other five one-stage detection models. In terms of individual model performance, the two-stage detection model of Cascade RCNN exhibited the best performance among the eight models, with *mAP*, *mAP50*, and *mAP75* values of 74.0%, 92.4%, and 85.0%, respectively. Among the five one-stage models on the GBDD1433-2023 dataset, ATSS performed the best, followed by the GFL model. Regarding the model’s running efficiency, as measured by the FPS metric, the two-stage detection models (average FPS value of 20.5) demonstrated lower efficiency compared to the one-stage detection models (average FPS value of 25.83). This indicates that two-stage models have higher detection accuracy but relatively lower running efficiency, while one-stage detection models have relatively lower detection accuracy but higher running efficiency.

Finally, we further compare the visualization effects of three representative models, Cascade RCNN, ATSS, and GFL on the GBDD1433-2023 dataset (Figure 6). From the visual comparison, it can be observed that Cascade RCNN achieves the best object detection performance. However, when faced with bird occlusion (Figure 6D), it fails to detect the entire bird. ATSS exhibits similar visualization effects to Cascade RCNN, but it also struggles to detect the complete bird when occluded (Figure 6D), albeit with a smaller detection area compared to Cascade RCNN. GFL shows relatively poor visualization results as it fails to detect the occluded bird (Figure 6D).

### 3.3. Model Robustness Analysis

The proportions of bird objects and the natural backgrounds they inhabit in real-world applications are complex and diverse, posing challenges for the application of object detection models. Evaluating the sensitivity of object detection models to different scenes and variations in bird object scales is of significant value in advancing the application of object detection models in the intelligent monitoring of wild birds.

To address this, we first trained the aforementioned eight object detection models using our self-built GBDD1433-2023 dataset, and we then evaluated the performance of these models using our self-built multi-modal bird object detection test dataset. We assessed the model performance using the accuracy rate metric, which is calculated as the number of bird images correctly detected divided by the total number of images. The experimental results showed that the two-stage object detection models (Faster RCNN, Cascade RCNN, and Libra RCNN) achieved a mean detection accuracy rate of over 90% in all 12 scene cases (Table 3). Among other things, the detection accuracy was 99%, even in the presence of interference from other animals. In comparison to the two-stage object detection models, the average detection accuracy rate of the one-stage object detection models (except YOLO-v3) was 87% across the 12 scene cases, indicating that the two-stage models exhibit stronger adaptability and anti-interference capabilities in different scenes compared to the one-stage models. Furthermore, the performance of the detection models was influenced by scene variations. In nine of the scene cases, such as grassland, farmland, mountain, and forest lands, the average detection accuracy rate of the eight detection models exceeded 90%, while in three scene cases including coastlines, riverbanks, and branches, the average detection accuracy rate decreased to 87.1%. Compared to grassland, farmland, mountain, and forest areas, the backgrounds of coastlines, riverbanks, and branches can be more complex. For instance, coastlines and riverbanks may have elements like ripples and reflections that could visually blur the shape or color of birds. Similarly, branches with leaves, fruits, or other elements can also pose visual challenges.

We further evaluated the performance of the eight object detection models under the nine bird object scales (25 × 25, 50 × 50, 75 × 75, 100 × 100, 125 × 125, 150 × 150, 175 × 175, 185 × 185, and 200 × 200). The experimental results (Table 4) showed that, similar to the previous results, the two-stage detection models generally outperformed the one-stage detection models in terms of detection performance. Moreover, the overall performance of the detection models improved as the bird object scale increased. For example, at a bird object scale of 25 × 25, the average detection accuracy rate of the eight models was 60.9%, while at a scale of 50 × 50, the average detection accuracy rate increased to 89.2%. When the bird object scale exceeded 75 × 75, the average detection accuracy rate of the eight models exceeded 91.8%. This indicates that the performance of the above eight object detection models could be improved in detecting small objects. For small object detection, the two-stage model has an average detection accuracy of 65.0 for 25 × 25 objects, which is significantly better than the 58.4 of the one-stage model. This indicates that the two-stage model generally performs better in small object detection. However, its larger model parameter size, to some extent, limits its practical application value.

### 3.4. Performance Comparison of Different Models in Bird Counting Tasks

Bird counting is an important task in bird field monitoring and surveys. Traditionally, this task has been carried out manually, which is time-consuming and labor-intensive. Evaluating the detection performance of object detection models for automatic bird counting tasks can contribute to the development of intelligent monitoring techniques for bird populations. We trained the aforementioned eight object detection models on the GBDD1433-2023 dataset and evaluated their detection performance using our self-constructed multi-object bird test dataset. Additionally, we used the counting accuracy metric (number of correctly detected birds/total number of observed birds) to characterize the model performance. The experimental results (Table 5) show that the average accuracy of the two-stage detection models was 62.1% in five different bird count scenarios (N = 2, N = 3, N = 4, N = 5, and N > 5), which is higher than the mean accuracy of the one-stage models (57.0%). Furthermore, as the number of birds increased, the counting accuracy of the eight detection models showed a downward trend. For example, when the bird count was N = 2, the counting accuracy of the eight models ranged from 74.1% to 77.2%; whereas for N > 5, the accuracy of the eight models decreased from 20.5% to 43.4%. Additionally, the visualized bird counting results of the three best-performing models in the bird counting task (Cascade RCNN, ATSS, and RetinaNet) also confirmed that the detection performance of the models was poorer when N > 5 (Figure 7). This clearly indicates that object detection models are not suitable for bird counting tasks when there are a large number of birds. We believe the main reason for this is that when the number of birds in a picture is too high, many bird objects are obscured due to factors such as bird movement, flock density, and shooting angles. As a result, the object detection network cannot accurately detect the objects.

### 3.5. Self-Adaptive Localization Distillation Experiment and Analysis

Due to the limitations of computational resources on edge devices, lightweight detection models are commonly used for deployment. To improve the performance of lightweight detection models, this paper proposes an self-adaptive localization distillation (SLD) framework. We selected four single-stage detection models, namely GFL, FCOS, ATSS, and RetinaNet to validate the effectiveness of SLD. In the experiment, we used ResNet101 as the teacher model’s backbone and ResNet50 as the backbone for the lightweight student model. We evaluated the impact of three knowledge distillation strategies (/: no distillation, LD: localization distillation, and SLD: self-adaptive localization distillation) on model performance when using the benchmark detection dataset COCO. The specific experimental results are shown in Table 6.

The experimental results indicate that, compared to the two previous strategies (no distillation and LD), the adoption of the SLD strategy improves the performances of the four single-stage detection models to some extent. On average, the SLD strategy increases the *mAP* of the student models by 2.5%, which is 0.4% higher than the LD strategy. In terms of individual model performance, the SLD strategy shows the largest performance improvement on RetinaNet, increasing its *mAP* value by 2.8%. This improvement surpasses the 2.5% improvement achieved by the LD strategy. This is mainly because RetinaNet has lower initial performance compared to the other three models, thus making it easier to achieve performance gains. Additionally, considering the differences in the *mAP50* and *mAP75* metrics among the four detection models, it is evident that the SLD strategy has a more pronounced effect on *mAP75*, which represents more precise localization. This phenomenon is particularly significant in the experiments with RetinaNet. For example, compared to LD, using SLD leads to no improvement in *mAP50* for the RetinaNet model, but it achieves a 0.5% performance improvement in the more fine-grained *mAP75*, thus resulting in an overall improvement in *mAP*. This indicates that, in the self-adaptive localization distillation strategy, it is effective to use real labels as the teacher during training when the student model encounters situations where the teacher model itself has poor recognition performance. This further indicates that SLD can help students choose more suitable teachers based on the current samples, thereby further improving the performance of the student model in precise detection (challenging samples). Overall, the aforementioned experimental results confirm the effectiveness of our designed adaptive localization distillation in improving the performance of lightweight detection models, thus surpassing traditional distillation methods.

## 4. Discussion

Birds, as important indicator species of ecosystems, provide crucial information about the health of ecological systems through their species and population changes. Additionally, data on bird species and population serve as fundamental information for research on bird ecology and biodiversity conservation. Traditionally, these data have been primarily collected through field surveys conducted by professional scientists and trained bird enthusiasts [47,48,49]. However, this data acquisition method is generally time-consuming and labor-intensive. With the development of computer vision and deep learning technologies, object detection techniques hold the potential to provide a low-cost and efficient means of data acquisition for bird surveys, thus enabling researchers to conveniently access key information on bird population dynamics, migration patterns, and habitat conditions [50,51,52].

However, the application of object detection technology in bird surveys and monitoring research faces challenges due to the lack of available large-scale bird object detection datasets and limited technical references for evaluating detection model performance and model lightweighting performance in bird detection tasks. To address these challenges, we conducted in-depth research on bird object detection dataset construction, detection model performance comparisons, and model lightweighting. We have successfully built the GBDD1433-2023 dataset, which is currently the most comprehensive bird object detection dataset worldwide as it achieves a detection accuracy metric *mAP50* of over 90% for the eight mainstream object detection models. This dataset not only provides professional bird basic datasets for researchers in the fields of deep learning and bird studies, but it also promotes the application of object detection technology in intelligent bird recognition, thus reducing the human and material resources required for researchers to construct bird object detection datasets from scratch.

There was a higher improvement in the performance of these eight object detection models on the GBDD1433-2023 datasets than that on the COCO dataset (Table 1 and Table 2). This can be explained due to the following reasons: (1) Although the GBDD1433-2023 dataset covers 1433 bird species, it primarily consists of a single-bird object class with high shape similarity among different bird species, thus making it easier for the models to learn shape features. (2) The COCO dataset includes more than 90 object classes with highly diverse appearances, which not only increases the difficulty for detection models to learn features for each category, but also leads to higher false detection rates and decreased detection performance. In summary, by constructing a manually annotated dataset specifically for bird object detection tasks, we can ensure that the dataset only contains relevant bird samples, thus reducing the influence of other species and background interference, as well as aiding in attaining a more accurate bird detection model performance.

By evaluating the sensitivity of the eight mainstream object detection models to scene variations and bird object scale changes, we found that the two-stage detection models outperformed single-stage detection models (Table 3 and Table 4). Furthermore, in terms of individual model performance, the two-stage Cascade RCNN model generally exhibited the best performance [53], while YOLOv3 and ATSS performed better among the single-stage detection models [54]. The superior performance of the Cascade RCNN model can be attributed to its multi-stage cascaded architecture, which combines the strengths of multiple two-stage models [53]. However, this model also has drawbacks, including a large number of model parameters (up to 265M) and low computational efficiency (20.5 f/s). The YOLOv3 model showed a good performance in scene variation experiments, which can be attributed to its operational strategy. YOLOv3 divides the image into multiple grids and predicts multiple bounding boxes and class probabilities for each grid, thus enabling location-sensitive detection [55]. Therefore, even if the bird background changes, YOLOv3 can accurately detect the objects by predicting their positions and classes within each grid, and this is achieved without being affected by the background. Additionally, the single-stage ATSS method performed well in multiple experiments due to the following reasons [43]: (1) The ATSS model adopts an anchor-free detection framework, and this can better adapt to objects of different sizes and shapes. (2) The ATSS model uses Focal Loss to alleviate the class imbalance problem, thus enabling better differentiation of difficult and easy samples, as well as improving the model’s accuracy. (3) The ATSS model employs a multi-scale feature fusion strategy, thus enabling for a better handling of objects with different scales, as well as in achieving superior capability in detecting small objects.

The proposed SLD framework provides an effective method for optimizing the performance of lightweight detection models. By introducing the concept of adaptive localization distillation, we are able to leverage the knowledge from the teacher model and the ground truth labels to guide the learning of the lightweight student model, thereby improving its performance and accuracy. This is of great significance for deploying efficient detection models on resource-constrained devices. In addition to adaptive localization distillation, combining other model compression and pruning techniques is also an effective approach through which to enhance the performance of lightweight models. Future research can explore the integration of adaptive localization distillation with other model compression techniques to further reduce the model’s parameters and computational complexity, thereby improving efficiency and performance. Overall, regarding the selection of object detection models for bird object detection and counting tasks, we provide the following recommendations:Two-stage object detection models (such as Cascade RCNN) exhibit high detection accuracy but slower runtime efficiency, making them suitable for scenarios that require high bird detection precision but not real-time detection. On the other hand, one-stage object detection models (like ATSS) have lower detection accuracy but faster runtime efficiency, making them suitable for applications such as real-time bird recognition and counting.The performance of object detection models is influenced by changes in application scenarios and the characteristics of the actual devices. When bird detection scenarios or devices result in the relatively weak performance of detection models, increasing the number of training samples of bird species can be considered.Small object detection remains a common challenge in the field of object detection. When a significant portion of bird objects in the application scenario are small, it is important to optimize the performance of object detection models. For example, improvements in network architecture, better data augmentation techniques, and multi-scale feature fusion can be explored to enhance the accuracy of small bird object detection.When the number of birds in the scene exceeds five, the counting accuracy of the object detection model is significantly affected by factors such as crowding and occlusion among the birds. It is recommended to consider using density estimation methods (such as CAP [56]) to address bird counting issues.

## 5. Conclusions

We have constructed the largest bird detection dataset to date, covering a wide range of common bird species globally. Utilizing this dataset, we have achieved a detection performance improvement of over 90% in terms of the mAP50 metric for eight mainstream object detection models that were specifically applied to bird detection tasks. Through a systematic evaluation of the current object detection models’ sensitivity to scene variations and changes in bird object scales, as well as their accuracy in bird counting tasks, we have not only confirmed the superior performance of two-stage detection models over single-stage detection models in bird detection tasks, but also discovered that object detection models are not suitable for bird counting tasks when the number of birds in images exceeds five. Our research results not only provide accurate and comprehensive data support for the field of bird detection, but also serve as a technical reference for users in selecting appropriate object detection models. Based on our research findings, when users prioritize detection accuracy over model computational efficiency, they can opt for two-stage detection models. Conversely, when users emphasize model runtime efficiency, particularly in the deployment of detection models on terminal devices, they can consider applying self-adaptive localization distillation to improve the performance of single-stage models. Given the vast number of bird species worldwide and the continuous emergence of new object detection technologies, we will continue to expand the scale of the bird detection dataset and evaluate the performance of additional object detection models in the future. These efforts will provide comprehensive data and technical support for bird intelligence recognition and monitoring research, thereby promoting the advancement of avian ecology conservation and biodiversity studies.

## Figures and Tables

**Figure 1 animals-13-02924-f001:**
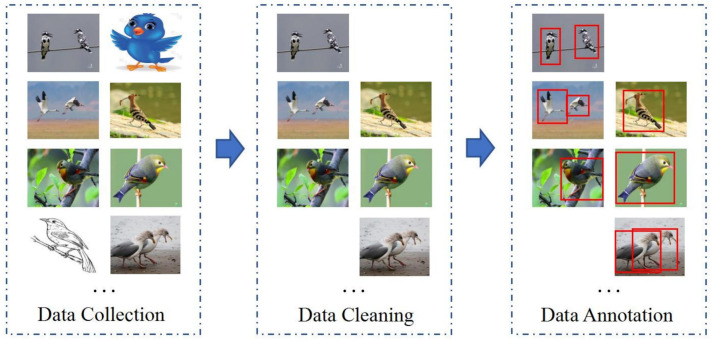
The flowchart on the construction of the bird object detection dataset (GBDD1433-2023).

**Figure 2 animals-13-02924-f002:**
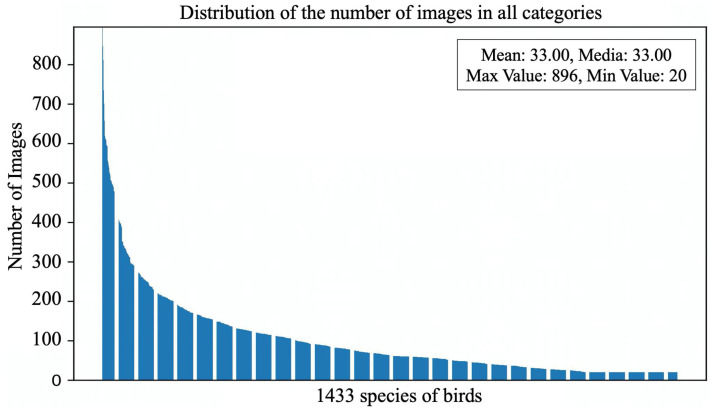
Overall distribution of the number of pictures of 1433 bird species in the GBDD1433-2023 dataset.

**Figure 3 animals-13-02924-f003:**
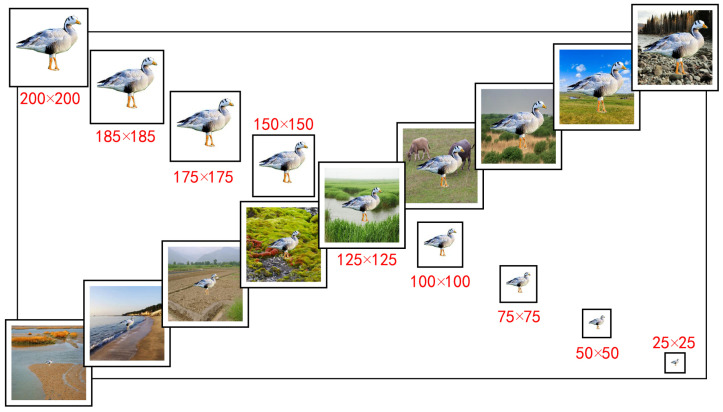
An example of a foreground bird image (bar-headed goose) shown at nine different sizes combined with nine background images.

**Figure 4 animals-13-02924-f004:**
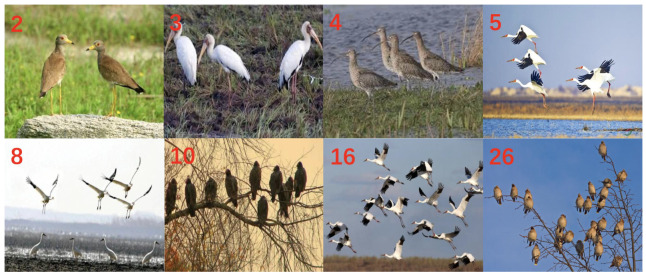
Examples of eight bird images with different bird numbers. The red numbers in each picture indicate the number of birds in that picture.

**Figure 5 animals-13-02924-f005:**
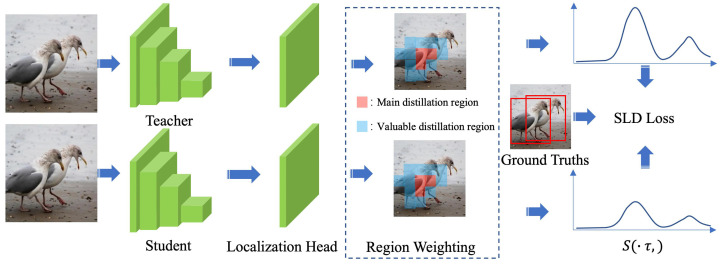
The overall training proccess of Self-Adaptive Localization Distillation (SLD).

**Figure 6 animals-13-02924-f006:**
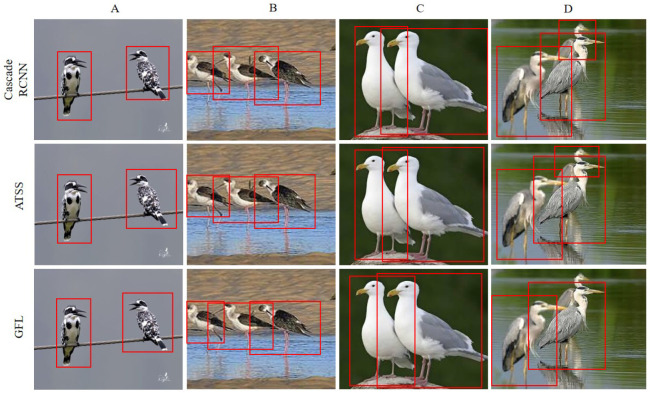
Visualized comparison of Cascade RCNN, ATSS, and GFL on the GBDD1433-2023. The subfigures of (**A**–**D**) represent the detection results of the models for four different scenarios.

**Figure 7 animals-13-02924-f007:**
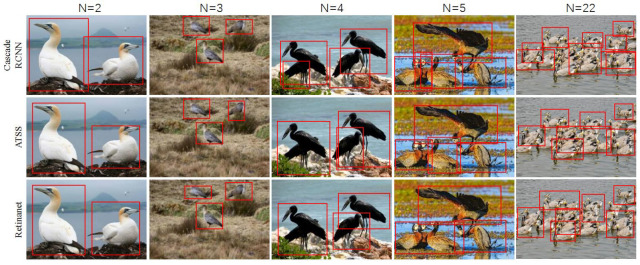
Visualized comparison of the three models of Cascade RCNN, ATSS, and RetinaNet on bird counting task.

**Table 1 animals-13-02924-t001:** Performance comparison of eight object detection models based on the COCO dataset. The bolded numbers represent optimal performance.

	Model	Backbone	mAP (%)	mAP50 (%)	mAP75 (%)	Model Size (M)
Two Stage	Faster RCNN	ResNet50	37.4	58.1	40.4	159.5
	Cascade RCNN	ResNet50	**40.3**	58.6	**44.0**	265.0
	Libra RCNN	ResNet50	38.3	**59.5**	41.9	160.5
One Stage	YOLOv3	ResNet50	39.1	56.9	41.9	175.9
	ATSS	ResNet50	39.4	57.6	42.8	123.5
	GFL	ResNet50	40.2	58.4	43.3	124.0
	FCOS	ResNet50	38.7	57.4	41.8	**123.5**
	RetinaNet	ResNet50	36.5	55.4	39.1	145.1

**Table 2 animals-13-02924-t002:** Performance comparison of the eight object detection models based on the GBDD1433-2023 dataset. The bolded numbers represent optimal performance.

	Model	Backbone	mAP (%)	mAP50 (%)	mAP75 (%)	FPS (f/s)
Two Stage	Faster RCNN	ResNet50	73.8	92.3	84.9	23.9
	Cascade RCNN	ResNet50	**74.0**	**92.4**	**85.0**	20.5
	Libra RCNN	ResNet50	73.4	92.1	83.9	23.5
One Stage	YOLOv3	ResNet50	70.8	91.6	82.1	30.1
	ATSS	ResNet50	73.2	91.7	84.4	29.8
	GFL	ResNet50	71.8	91.3	83.5	28.9
	FCOS	ResNet50	68.5	90.6	80.4	27.3
	RetinaNet	ResNet50	72.5	91.0	82.9	25.8

**Table 3 animals-13-02924-t003:** The Impact of Scene Changes on the Performances of the Eight Object Detection Models.

Model	Grassland	Farmland	Mountain Peak	Forest	Shrubland	Seaside	Tree Branch	River Bank	Tundra	Reed Swamp	Mudflat	Interference
Faster RCNN	97.6	96.6	98.9	91.1	97.5	91.6	91.7	94.3	91.7	96.8	98.0	99.5
Cascade RCNN	96.9	92.4	94.1	92.2	95.3	98.1	86.8	90.8	89.8	93.6	93.6	99.6
Libra RCNN	94.5	95.3	90.2	90.3	97.2	94,1	90.1	92.3	93.2	93.2	91.4	99.2
YOLOv3	93.0	96.2	94.3	95.3	96.3	94.9	91.4	94.0	95.0	95.9	95.9	99.2
ATSS	86.0	84.6	91.3	89.2	87.4	85.5	89.3	80.6	90.9	93.1	90.0	89.3
GFL	86.4	87.7	77.8	85.8	79.7	68.5	82.3	73.7	83.4	90.1	80.1	82.2
FCOS	87.4	88.0	85.9	87.6	84.4	79.1	84.8	80.6	85.8	91.4	83.8	78.2
RetinaNet	88.4	88.2	93.9	89.3	89.0	89.6	87.2	87.4	90.1	92.7	87.5	74.1
Mean	91.3	91.1	90.8	90.1	90.8	86.8	87.9	86.7	90.0	93.4	90.0	90.2

**Table 4 animals-13-02924-t004:** The impact of bird target scale variation on the performances of the eight object detection models.

Model	25 × 25	50 × 50	75 × 75	100 × 100	125 × 125	150 × 150	175 × 175	185 × 185	200 × 200
Faster RCNN	62.9	95.3	96.5	98.1	98.2	99.2	99.3	99.2	99.4
Cascade RCNN	67.3	96.3	96.9	98.4	98.6	99.1	99.1	99.3	99.7
Libra RCNN	65.1	96.9	96.7	98.1	98.7	99.1	99.2	99.7	99.6
YOLOv3	60.5	96.2	97.4	97.5	97.8	97.9	98.7	98.7	99.2
ATSS	63.4	86.8	88.2	88.4	88.2	89.1	93.7	95.1	94.3
GFL	56.1	71.5	79.6	90.3	95.2	97.1	96.4	95.8	94.3
FCOS	58.3	83.9	88.5	93.9	96.5	97.5	97.55	97.3	96.8
RetinaNet	53.6	86.0	90.3	93.8	98.0	99.7	99.7	99.7	99.3
Mean	60.9	89.2	91.8	94.8	96.5	97.3	98.0	98.0	97.0

**Table 5 animals-13-02924-t005:** Counting Accuracy of the Eight Object Detection Models for Different Bird Quantities.

	Model	Mean (%)	N = 2	N = 3	N = 4	N = 5	N > 5
Two-Stage	Faster RCNN	61.1	75.6	74.6	70.2	62.6	23.9
	Cascade RCNN	63.0	77.2	75.8	71.8	63.9	20.5
	Libra RCNN	62.3	74.1	73.4	69.6	63.3	23.5
One-Stage	YOLOv3	55.8	70.5	69.9	66.7	60.8	38.9
	ATSS	59.3	75.7	73.5	66.3	63.4	43.4
	GFL	54.9	71.5	70.1	66.1	61.3	37.9
	FCOS	56.8	74.5	72.9	67.1	60.2	38.3
	RetinaNet	58.4	75.2	72.7	65.4	62.9	44.1

**Table 6 animals-13-02924-t006:** LD vs. SLD ablation experiments on the COCO dataset. In the table, R101 and R50 represent the backbone network ResNet101 and ResNet50, respectively. The bolded numbers represent the optimal performance.

Teacher	Student	/	LD	SLD (Ours)	mAP (%)	mAP50 (%)	mAP75 (%)
ATSS-R101	ATSS-R50	✓			39.4	57.6	42.8
ATSS-R101	ATSS-R50		✓		41.6	59.3	45.3
ATSS-R101	ATSS-R50			✓	**42.1**	**59.5**	**45.8**
GFL-R101	GFL-R50	✓			40.2	58.4	43.3
GFL-R101	GFL-R50		✓		42.1	60.3	45.6
GFL-R101	GFL-R50			✓	**42.4**	**60.5**	**46.1**
FCOS-R101	FCOS-R50	✓			38.7	57.4	41.8
FCOS-R101	FCOS-R50		✓		40.6	58.4	44.1
FCOS-R101	FCOS-R50			✓	**41.0**	**58.5**	**44.5**
RetinaNet-R101	RetinaNet-R50	✓			36.5	55.4	39.1
RetinaNet-R101	RetinaNet-R50		✓		39.0	**56.4**	42.4
RetinaNet-R101	RetinaNet-R50			✓	**39.3**	**56.4**	**42.9**

## Data Availability

Our self-constructed GBDD1433-2023 bird object detection dataset is available for download at https://github.com/yangyangtiaoguo/GBDD-1433-2023, accessed on 1 August 2023.

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
