# Peer review of "Bird Object Detection: Dataset Construction, Model Performance Evaluation, and Model Lightweighting"

_animals, 2023, doi:10.3390/ani13182924_

Round 1
Reviewer 1 Report
This is a comprehensive study focusing on bird detection. The authors have assembled the largest known bird object detection dataset. They compared the performances of eight mainstream detection models for bird detection tasks and proposed a feasible approach for model lightweighting specific to bird object detection. However, there are some issues that need to be addressed before publication.
Line 45: Please include the reference (Yang, Xiao, et al. "A deep learning model for detecting cage-free hens on the litter floor." Animals 12.15 (2022): 1983) to highlight a use case of deep learning in bird detection.
Line 51: Given the advancements since YOLOv3, it would be prudent to reference more recent YOLO algorithms as YOLOv3 is now considered outdated.
Line 174: Could you provide a clearer description of the data cleaning and processing methods you utilized?
Results section: I noticed that you employed mAP, mAP50, and mAP75 metrics to assess model performance. However, the relevant equations for these calculations are missing. Kindly add this information to the materials and methods section.
Line 216: Your statement indicates that the two constructed test datasets, one of which is the global bird object detection dataset, don't overlap. Yet, in Figure 7 where N=22, there's an evident overlap of birds. This raises questions about the distinctiveness of your test datasets. Is there another test dataset that wasn't discussed in the paper?
Author Response
Reviewer 1
1、Line 45: Please include the reference (Yang, Xiao, et al. "A deep learning model for detecting cage-free hens on the litter floor." Animals 12.15 (2022): 1983) to highlight a use case of deep learning in bird detection.
Answer: Good suggestion and thanks. We found that this is a very valuable reference on deep learning in bird detection, and we have cited it in the appropriate place in our paper (Line 44, refer 12).
2、Line 51: Given the advancements since YOLOv3, it would be prudent to reference more recent YOLO algorithms as YOLOv3 is now considered outdated.
Answer: Good suggestion and thanks. You are absolutely right. Currently, the YOLO series has been updated to YOLOv8, and YOLOv3 is a relatively outdated work. Moreover, experimenting with the newer YOLO series is not difficult. However, the focus of this paper is to analyze the most representative methods, not just to focus on the latest YOLO technology. YOLOv3 has been widely used and is one of the most classic methods in the YOLO series, having made a profound impact at its time. Although the subsequent YOLO series have better performances, they are improved by borrowing components or designs from other different models. In terms of comparing the performances of detection models with other mainstream models, this actually increases the unfairness of performance comparisons. Therefore, we think that although YOLOv3 is an outdated method, it is relatively representative and has more analytical value for comparing different types of methods.
3、Line 174: Could you provide a clearer description of the data cleaning and processing methods you utilized?
Answer: Good suggestion. Based on your suggestions, we reorganized the relevant information and provided a more detailed and clear description of the data cleaning and annotation methods which were used for the dataset construction process. (Lines: 169-177)
4、I noticed that you employed mAP, mAP50, and mAP75 metrics to assess model performance. However, the relevant equations for these calculations are missing. Kindly add this information to the materials and methods section.
Answer: Good suggestion and thanks. In previous manuscripts, we ignored the possible lack of understanding of the evaluation metric of mAP for readers not in the AI field. We have supplemented a detailed description of the mAP, mAP50, and mAP75 at the corresponding position. For example, “…The detection performance of the object detection model is evaluated using the Mean Average Precision (mAP) metric. mAP represents the average area under the precision-recall curve for all categories at a specified Intersection over Union (IOU) threshold. IOU refers to the overlap ratio between the detection box and the ground truth box, which can be used to measure the degree of match between the detection result and the real object. Typically, the higher the mAPx value at a given IOU threshold x (ranging from 0 to 1), the better the model's detection performance. To calculate mAP, one must first compute the Precision and Recall values for each category and then plot the Precision-Recall curve. For each category, the area under its Precision-Recall curve is calculated to obtain the AP for that category, which measures the detection performance of a single category. The average of the APs for all categories results in the mAP, which is the mean value of AP for all categories. The formulas for computing Precision and Recall are as follows…” (Lines: 396-406)
5、Line 216: Your statement indicates that the two constructed test datasets, one of which is the global bird object detection dataset, don't overlap. Yet, in Figure 7 where N=22, there's an evident overlap of birds. This raises questions about the distinctiveness of your test datasets. Is there another test dataset that wasn't discussed in the paper?
Answer: Thank you very much for your meticulous suggestion. The ambiguity may have arisen from the unclear phrasing in the sentence on line 216. What we intended to convey is that the samples in the two constructed test datasets do not overlap with the images in the training set, rather than suggesting that the bird targets within the images do not overlap. We have revised the sentence to ensure its clarity (Lines: 213-215).

Reviewer 2 Report
The authors built up an extensive dataset for identifying birds in the wild. They propose an adaptive object detection distillation method that avoids the propagation of erroneous knowledge, further improving the detection performance of lightweight detection models.
I found the manuscript interesting as the authors point out the benefits of using different models for better accuracy.
Introducing the landscape in bird identification increases the possibility of successful identification, therefore improving the target.
I just suggest some review in the text attached here.

I suggested minor corrections in the attached text.
Author Response
1、Some grammatical problems in the manuscript as listed below:
(1)“…We systematically evaluate the performance of mainstream two-stage and...”.;(2)“To achieve this, we introduce the confidence ...”
Answer: Thank you for pointing out grammatical errors in the paper, we have changed both the two verbs from present tense to past tense (Line:147 and Line:344)
2. Use Arabic numerals in the headings of Tables 1-5. For instance, “…Performance comparison of 8 object detection models based on the COCO dataset…” (Table 1.)
Answer: Thank you for your meticulous suggestion. Following your suggestion, we have replaced all Arabic numeral 8 with "eight" in the titles of tables 1-5 in the revised version(Lines: 437, 465, 507, 508 and 530).
3. However, it is not feasible to evaluate the performance of all object detection models in bird object detection tasks.
Answer: Thank you for pointing out that there is an expression problem with this sentence. In the revised version, we have rewritten the sentence as “Considering a large number of published object detection algorithms, comparing the performances among all methods is beyond the scope of this paper.”
4. “…However, each image in this dataset only contains a single bird, which does not meet the requirements of bird counting or multi-bird recognition tasks...”. Please refer to this as a limitation of the previous model.
Answer: Good suggestion. Based on your suggestions, we revised the sentence to emphasize the limitations posed by the absence of multiple birds in a single image (Lines: 96-98).
5." …the selected models must be recognized by the academic community as having excellent performance in object detection, considering the constant emergence of new methods and techniques in the field…" Avoid using superlative words in scientific text.
Answer:Good suggestion. Regarding the issue of superlative words, we modified the descriptions in the related sentences. For instance, “…given the continual emergence of novel methods and technologies in the field, the chosen model must be academically recognized as the state-of-the-art at that time…” (Line:226)
6. "...generalized softmax function S (·, τ) to obtain.. loss H (·) ..." Which is this variable? or boundary? Please clarify it.
Answer: Thank you for your meticulous suggestion. Concerning the unclear description of the S (·, τ) and H (·) formula, we have added a detailed description for this formula in the revised manuscript (Lines: 327-332).
7." …The confidence parameter C can adjust the magnitude of the loss values generated by the student and teacher models during distillation. A higher value of C encourages the student model to lean towards learning from the teacher model, while a lower value does the opposite...”. Do you have a reference for this step taken?
Answer: Thank you for your meticulous suggestion. As for your suggestion to add a reference, we have included a paper titled "Adaptive Multi-Teacher Multi-Level Knowledge Distillation" as a reference in the 358 line.
8." ... In nine scene cases such as grassland, farmland, mountain and forest, the average detection accuracy rate of the eight detection models exceeded 90%, while in three scene cases including coastline, riverbank and branch, the average detection accuracy rate decreased to 87.1%...". Do you have any idea why this happened? Even though is a small reduction in accuracy, it would be interesting to know the reason for the reduction.
Answer: Good question. About the reduced performance of bird detection in the scenarios of coastlines, riverbanks, and branches, we think that there are mainly the following two reasons: (1). Background Complexity: Compared to grasslands, farmlands, mountains, and forests, the backgrounds of coastlines, riverbanks, and branches might be more complex. For instance, coastlines and riverbanks may have elements like ripples and reflections, which might blur the shape or color of the birds visually. Similarly, branches containing leaves, fruits, or other elements can also pose visual challenges. (2). Lighting Variation: The lighting conditions in places like coastlines and riverbanks might change frequently, possibly due to reflections or the positioning of the terrain. In scenes involving branches, the shadows of the trees might affect visual clarity. In future, we will consider how to further design experiments to confirm our hypothesis.
9.“… This clearly indicates that object detection models are not suitable for bird counting tasks when there are a large number of birds…” On line 156 you propose a model better the one available, based on the fact the previous model is not efficient to register more than five birds. It would be helpful to enhance that you also could not detect when birds are more than five in the flock.
Answer: Good question. In fact, we do not transform the structure of the one-stage object detection model, we just use the knowledge distillation technique to allow a small model to assimilate some of the learning capabilities of the larger model. However, this capability enhancement is limited and will not change the basic characteristics of the model. Therefore, when detecting images containing more than 5 birds, the performance of this improved model will still be similar to the teacher model (larger model), and still perform poorly on small objects. For the improved small model, although its detection performance is close to that of the large model, it operates with improved efficiency and is suitable for deployment to field monitoring equipment.
10.“…When the number of birds in a scene becomes excessively high, the counting accuracy of object detection models is affected by factors like crowdedness and occlusion among birds...”
Answer: Thank you for reminding us of the unscientific and inaccurate expression. We have revised the sentence. The new sentence is “When the number of birds in a scene exceeds 5”.
11. For references, please check them.
Answer: Thank you for your meticulous suggestion. In the revised manuscript, the abbreviations of all journal names in our references were checked and several incorrect abbreviations were corrected. All modifications are highlighted in red in the revised manuscript.

Reviewer 3 Report
In this study, we not only compare the lack of bird object detection datasets and the performance of object detection models by creating the largest known bird object detection dataset but also by comparing the performance of eight mainstream detection models on bird object detection tasks. A feasible approach for model mitigation in bird object detection is then proposed. On the bird dataset, the two-stage object detection models (Faster R-CNN, Cascade R-CNN, and Libra R-CNN) achieved an average precision (mAP) of 73.7%. Their performance was superior to five single-stage object detection models (YOLOV3, ATSS, GFL, FCOS and RetinaNet). The performance differences of the models were also evaluated on bird counting tasks, and the bird counting accuracy of the object detection models ranged from 60.8% to 77.2%.
In this sense, it is clear that this study will contribute to the literature in this field. Although it uses known data processing methods, it will provide important contributions in terms of application. The results of the study are satisfactory. However, the Abstract needs to be simplified. I also suggest that a flowchart section should be added to the paper. Table 3-6 needs to be explained in detail.
Author Response
1.The Abstract needs to be simplified.
Answer: Good suggestion and thanks. As required by the Animals journal format, there are two for the research brief: (1) simple summary and (2) abstract. In the original manuscript, we refined the abstract and generated a simple summary (Lines:1~8). In the revised manuscript, we have compressed the abstract appropriately according to your comments (Lines:9~25)
2. I also suggest that a flowchart section should be added to the paper.
Answer: Good suggestion and thanks. We have added an overall flow chart section to the text (Lines: 142). Our overall flow is shown below:
3. Table 3-6 needs to be explained in detail.
Answer: Thank you for your careful advice. According to your suggestions, we have provided a more detailed description of the relevant tables. For example, in revised manuscript, “…then experimental results showed that the two-stage object detection models (Faster RCNN, Cascade RCNN, Libra RCNN) achieved a detection mean accuracy rate of over 90% in all 12 scene cases (Table 3). Among other things, the detection accuracy was 99% even in the presence of interference from other animals. In comparison to the two-stage object detection models, the average detection accuracy rate of the one-stage object detection models (except YOLO-v3) was 87% across the 12 scene cases, indicating that the two-stage models exhibit stronger adaptability and anti-interference capabilities in different scenes compared to the one-stage models. Furthermore, the performance of the detection models was influenced by scene variations. In nine scene cases such as grassland, farmland, mountain and forest, the average detection accuracy rate of the eight detection models exceeded 90%, while in three scene cases including coastline, riverbank and branch, the average detection accuracy rate decreased to 87.1%. Compared to grassland, farmland, mountain, and forest, the backgrounds of coastline, riverbank, and branch can be more complex. For instance, coastline and riverbanks may have elements like ripples and reflections, which could visually blur the shape or color of birds. Similarly, branch with leaves, fruits, or other elements can also pose visual challenges…” (Lines:475~491).
In additionally, we also corrected some wording errors. Table3 (Lines: 476-492), Table4 (Lines: 497-507), Table5 (Lines: 522-530), and Table6 (Lines: 558-562).

Round 2
Reviewer 1 Report
Thanks for your reply. the paper is ready to be published now.